

# How mangabey molar form differs under routine vs. fallback hard-object feeding regimes

Debbie Guatelli-Steinberg[1,2], Cameron Renteria[3,4], Jack R. Grimm[3], Izabela Maeret Carpenter[3], Dwayne D. Arola[3,4,5,6] and W. Scott McGraw[1]

[1] Department of Anthropology, The Ohio State University, Columbus, OH, United States of America
[2] School of Anthropology and Conservation, University of Kent, Canterbury, Kent, United Kingdom
[3] Department of Materials Science and Engineering, University of Washington, Seattle, WA, United States of America
[4] Department of Oral Health Sciences, School of Dentistry, University of Washington, Seattle, WA, United States of America
[5] Department of Mechanical Engineering, University of Washington, Seattle, WA, United States of America
[6] Department of Restorative Dentistry, School of Dentistry, University of Washington, Seattle, WA, United States of America

Corresponding author
Debbie Guatelli-Steinberg,
guatelli-steinberg.1@osu.edu

## ABSTRACT

**Background**. Components of diet known as fallback foods are argued to be critical in shaping primate dental anatomy. Such foods of low(er) nutritional quality are often non-preferred, mechanically challenging resources that species resort to during ecological crunch periods. An oft-cited example of the importance of dietary fallbacks in shaping primate anatomy is the grey-cheeked mangabey *Lophocebus albigena*. This species relies upon hard seeds only when softer, preferred resources are not available, a fact which has been linked to its thick dental enamel. Another mangabey species with thick enamel, the sooty mangabey *Cercocebus atys*, processes a mechanically challenging food year-round. That the two mangabey species are both thickly-enameled suggests that both fallback and routine consumption of hard foods are associated with the same anatomical feature, complicating interpretations of thick enamel in the fossil record. We anticipated that aspects of enamel other than its thickness might differ between *Cercocebus atys* and *Lophocebus albigena*. We hypothesized that to function adequately under a dietary regime of routine hard-object feeding, the molars of *Cercocebus atys* would be more fracture and wear resistant than those of *Lophocebus albigena*.

**Methods**. Here we investigated critical fracture loads, nanomechanical properties of enamel, and enamel decussation in *Cercocebus atys* and *Lophocebus albigena*. Molars of *Cercopithecus*, a genus not associated with hard-object feeding, were included for comparison. Critical loads were estimated using measurements from 2D µCT slices of upper and lower molars. Nanomechanical properties (by nanoindentation) and decussation of enamel prisms (by SEM-imaging) in trigon basins of one upper second molar per taxon were compared.

**Results**. Protocone and protoconid critical fracture loads were significantly greater in *Cercocebus atys* than *Lophocebus albigena* and greater in both than in *Cercopithecus*. Elastic modulus, hardness, and elasticity index in most regions of the crown were greater in *Cercocebus atys* than in the other two taxa, with the greatest difference in the outer enamel. All taxa had decussated enamel, but that of *Cercocebus atys* uniquely exhibited a bundle of transversely oriented prisms cervical to the radial

enamel. Quantitative comparison of in-plane and out-of-plane prism angles suggests that decussation in trigon basin enamel is more complex in *Cercocebus atys* than it is in either *Lophocebus albigena* or *Cercopithecus cephus*. These findings suggest that *Cercocebus atys* molars are more fracture and wear resistant than those of *Lophocebus albigena* and *Cercopithecus*. Recognition of these differences between *Cercocebus atys* and *Lophocebus albigena* molars sharpens our understanding of associations between hard-object feeding and dental anatomy under conditions of routine vs. fallback hard-object feeding and provides a basis for dietary inference in fossil primates, including hominins.

## INTRODUCTION

Two mangabey species have figured prominently in discussions of hard-object feeding and enamel thickness in fossil hominins (*Grine et al., 2006*; *Ungar, Grine & Teaford, 2008*; *Daegling et al., 2011*; *McGraw, Vick & Daegling, 2014*; *Ungar, 2017*). The grey-cheeked mangabey *Lophocebus albigena* relies on hard seeds as a dietary fallback—when softer, preferred resources are not available (*Lambert et al., 2004*). Reliance on hard foods as critical fallbacks has been linked to thickened dental enamel in this species (*Lambert et al., 2004*). By contrast, the similarly thickly enameled sooty mangabey, *Cercocebus atys* of the Taï Forest, Ivory Coast, processes a mechanically challenging food, *Sacoglottis gabonensis*, year-round (*McGraw, Vick & Daegling, 2014*; *McGraw, Pampush & Daegling, 2012*). Seeds of this plant species are protected by casings twice as hard as cherry pits (*Daegling et al., 2011*). Fallback and routine consumption of hard foods are thus both associated with thick enamel, complicating dietary interpretation in fossil primates.

Despite both species being thickly-enameled, their molars have recently been shown to differ in three important ways (*Guatelli-Steinberg et al., 2022*). First, they differ in absolute crown strength (ACS), a metric based on tooth size and absolute enamel thickness that reflects fracture resistance (*Schwartz, McGrosky & Strait, 2020*). Second, they differ in the proportional thickness of enamel in their occlusal basins. Proportionally thicker enamel in occlusal basins affords greater resistance to fracture and can forestall wear-related dentine exposure in this crown region. Third, flare of the two mangabey species' molar sidewalls differs; greater flare improves buttressing against laterally directed chewing forces (*Singleton, 2003*; *Macho & Shimizu, 2009*). *Cercocebus atys* molars have greater ACS, proportionally thicker occlusal basin enamel, and greater flare than those of *Lophocebus albigena*, suggesting that routine and fallback hard-object feeding are not associated with identical molar form (*Guatelli-Steinberg et al., 2022*). Here, we further explore molar form in these species and a sample of *Cercopithecus*, aiming to better understand how molar form differs under conditions of fallback *vs.* routine hard-object feeding.

Although measurements of the mechanical properties of *Cercocebus atys* (*Daegling et al., 2011*) and *Lophocebus albigena* (*Lambert et al., 2004*) foods are not directly comparable, the

two species are known to differ in both oral processing behavior as well as the frequency with which they masticate hard foods (*Daegling et al., 2011*; *McGraw, Vick & Daegling, 2011*). *Cercocebus atys* uses its incisors to remove any adherents to *Sacoglottis gabonensis* seed casings and may attempt to puncture them with its anterior dentition. However, to access nuts within *Sacoglottis gabonensis*, *Cercocebus atys* typically places the large seed casings on its post-canine teeth where they are shattered with a powerful isometric bite (*Daegling et al., 2011*; *McGraw, Vick & Daegling, 2011*). *Lophocebus* species are thought to use their anterior teeth to a greater extent than do species of *Cercocebus* (*Daegling & McGraw, 2007*), consistent with the difference between the two genera in mandibular corpus depth (*Daegling & McGraw, 2007*). The deeper mandibular corpora of *Lophocebus* are more resistant to parasagittal bending forces produced during powerful incision than are the shallow mandibular corpora of *Cercocebus* (*Daegling & McGraw, 2007*). Compared to *Lophocebus*, *Cercocebus'* greater reliance on its molars in food processing as well as its more frequent consumption of hard-object foods would expose its molars to greater opportunity for fracture and increase their risk of fatigue failure.

Based on these considerations, we first build on a previous analysis of ACS in the two mangabey species (*Guatelli-Steinberg et al., 2022*) to ask if cusps directly involved in crushing and grinding (Phase II of the chewing cycle) are differentially strengthened in *Cercocebus atys* compared to those of *Lophocebus albigena*. For upper molars, the cusps most directly involved in Phase II are lingual cusps; for lower molars, these are buccal cusps (*Kay, 1975*). A shorthand we adopt is referring to upper molar lingual cusps and lower molar buccal cusp as "functional cusps" in recognition of their involvement in Phase II chewing, which is reflected in greater lateral wall enamel thickness of functional *vs.* non-functional cusps (*Schwartz, 2000*). Because of the frequency with which *Cercocebus atys'* functional cusps are loaded, we hypothesized that differences in the ability of *Cercocebus atys* and *Lophocebus albigena* molar cusps to resist fracture would be most pronounced in their functional cusps.

Then, to further investigate the dietary signal of proportionally thicker enamel in occlusal basins of *Cercocebus atys*, we compared nanomechanical properties and decussation complexity of trigon basin enamel in one molar each of *Cercocebus atys, Lophocebus albigena*, and, for additional context, *Cercopithecus cephus*, a species not known to consume hard foods (*Gautier-Hion, 1980*; *Gautier-Hion, Gautier & Quris, 1981*; *Tutin et al., 1997*). We chose to analyze the trigon basin because it is the major crushing basin of the upper molars (*Butler, 1972*).

We used nanoindentation to examine elastic modulus (E), the resistance of a material to reversible (*i.e.,* elastic) deformation, hardness (H), the resistance of a material to irreversible (plastic) deformation, and the ratio between the two (H/E), known as the elasticity index (*Leyland & Matthews, 2000*; *Labonte, Lenz & Oyen, 2017*). This comparison was made by one co-author (C.R.), who was blind to the species-identity of the molars. Although harder enamel is thought to be more wear-resistant than softer enamel (*Constantino et al., 2012*), differences in enamel hardness across a broad range of primate species were not found to correlate with dietary variation (*Constantino et al., 2012*). Here, in addition to enamel hardness, we analyzed the elasticity index, which may be a more accurate indicator

of wear resistance (*Leyland & Matthews, 2000*; *Labonte, Lenz & Oyen, 2017*). Given that *Cercocebus atys* routinely processes grit-laden foods from the forest floor (*Geissler, Daegling & McGraw, 2018*) and that mastication of hard foods can also cause wear (*Teaford & Oyen, 1989*), we hypothesized that *Cercocebus atys* would have the greatest elasticity index of the three species.

With respect to enamel decussation, *i.e.,* the crisscrossing of prisms in enamel, we assessed "enamel complexity", defined as "any microanatomical feature of dental enamel that increases the heterogeneity of enamel crystallite orientations" (*Hogg & Elokda, 2021*). The greater the heterogeneity of enamel prism orientations, the more difficult it is for cracks to propagate along prism boundaries (*Hogg & Elokda, 2021*; *Bajaj & Arola, 2009*). Decussation appears to be more complex in the enamel of mammalian species that experience high loading forces on their teeth (*Hogg & Elokda, 2021*). For example, marked enamel complexity in the canines of robust capuchins has been linked (*Hogg & Elokda, 2021*) to the high bite forces this species uses to process hard foods (*Wright, 2005*; *Alfaro, Silva Jr & Rylands, 2012*). Although complex decussating enamel resists crack propagation, it is not as wear-resistant as enamel in which prisms are arranged in parallel with the direction of abrasion (*Rensberger, 2000*). Thus, there is a prism orientation trade-off in enamel in terms of fracture *vs.* wear resistance. In most mammals, prisms tend to run parallel to one another in the outer enamel where they function to resist wear, while they are more decussated in the enamel that lies beneath this outer region, where they act to resist fracture (*Rensberger, 2000*).

We expected *Cercocebus atys*, like the other taxa in our study, to exhibit enamel with parallel prism orientation in the outer enamel of its trigon basin, conferring resistance to wear. However, we hypothesized that the underlying decussated enamel of its trigon basin would be more complex than it is in the other taxa. We expected greater complexity in *Cercocebus atys* not only because it processes hard food objects on its molars more frequently than do *Lophocebus albigena* and *Cercopithecus cephus*, but also because it has the highest estimated bite force among 23 primate species to which it has been compared (*Deutsch et al., 2020*), including significantly larger-bodied mandrills. Thus, our previously-stated expectation of greater wear-resistance in *Cercocebus atys* molars relates not to the arrangement of their enamel prisms, which we hypothesized would exhibit greater decussation complexity, but to the nanomechanical properties of their enamel.

## MATERIAL AND METHODS

### Critical margin fracture loads

$P_{MF}$ values, the critical loads necessary for margin fractures to propagate to crown failure (*Schwartz, McGrosky & Strait, 2020*), were calculated from measurements taken on 2D buccal-lingual μCT slices through mesial cusps. Margin fractures begin at the cervix of the crown and travel through the lateral enamel towards the cusp. We did not calculate critical loads for radial fractures, which start in the cusp (*Schwartz, McGrosky & Strait, 2020*), because most of our sample had some degree of wear at cusp tips.

*Sample* and μ*CT scanning*. For estimating critical fracture loads, our sample consisted of *Cercocebus atys* specimens from the Ivory Coast's Taï Forest, *Lophocebus albigena* specimens

from near Makoua, Republic of Congo, and specimens of *Cercopithecus* species from the Taï Forest (*C. campbelli* and *C. petaurista*) and Republic of Congo (*C. cephus*). The specimens used in this study are the dental remains of animals that were found dead (*e.g.*, natural causes, bushmeat). None of the monkeys whose remains were included in this study required an export permit, as they are CITES Appendix II taxa (Ivory Coast Field Permit for Exportation 2352).

Feeding data on *Cercocebus atys* at Taï indicate this species consumes hard foods year-round (*McGraw, Vick & Daegling, 2011*; *McGraw, Vick & Daegling, 2014*; *McGraw, Pampush & Daegling, 2012*), while those on *Lophocebus albigena* indicate the species prefers fruit but switches to hard seeds seasonally (*Lambert et al., 2004*; *Ham, 1994*; *Brugiere et al., 2002*; *Poulsen, Clark & Smith, 2001*; *Poulsen et al., 2002*; *Waser, 1984*; *Olupot et al., 1997*). For instance, at Lope, Gabon, *Lophocebus albigena* consume seeds *Pentaclethera macrophylla* (*Deutsch et al., 2020*), which are enclosed in tough, hard pods (*McGraw et al., 2016*). The guenon species in our study are not known to consume hard foods (*Buzzard, 2006*; *Buzzard, 2004*; *McGraw & Zuberbuhler, 2007*).

The full sample for margin fracture critical loads estimation is given in DataSet S1. For each antimeric pair of molars, the molar with least wear was chosen for scanning. We did not include sex in our analysis because information about sex was only known for a portion of our sample. Variation in molar form by sex is therefore not accounted for in this study. All teeth were manually extracted from the jaws of deceased animals (see above).

Scans were made with a Bruker Skyscan 1172 High Resolution Ex Vivo 3D X-ray Tomography Scanner (in the Do-Gyoon Kim Laboratory at the OSU College of Dentistry). Most of our scanning was done at 22 μm (with a few specimens at 13 μm). We used N.Recon v1.7.4.2 to process raw output into a TIFF format. Using Dragonfly v.2021.1.0.977, each three-dimensional digital rendering was virtually "sliced" along a bucco-lingual plane through the dentine horns of its buccal and lingual mesial cusps and perpendicular to its cervical margin. Virtual sections were saved as TIFFs, which were imported into Adobe Photoshop.

### Measurements

Figure 1 depicts measurement reference lines. AET (Average Enamel Thickness) was calculated as the area of the enamel cap divided by the length of the EDJ (Enamel-Dentine Junction) (*Martin, 1985*). Maximum lateral wall enamel thickness was measured at the widest point between the EDJ and OES (Outer Enamel Surface) along a line perpendicular to the EDJ (*Schwartz, McGrosky & Strait, 2020*; *Spoor, Zonneveld & Macho, 1993*; *Ulhaas, Henke & Rothe, 1999*; *Kono, Suwa & Tanijiri, 2002*; *Suwa & Kono, 2005*). BCD is the bicervical diameter of the crown.

### Crown reconstruction

Worn crowns were reconstructed based on recommendations given in *O'Hara & Guatelli-Steinberg (2021)*. *O'Hara & Guatelli-Steinberg (2021)* found that when using either the Profile (*Grine & Martin, 1988*; *Smith et al., 2011*; *Smith et al., 2012*) or Pen Tool reconstruction methods (*Saunders et al., 2007*; *O'Hara et al., 2019*), accurate AET

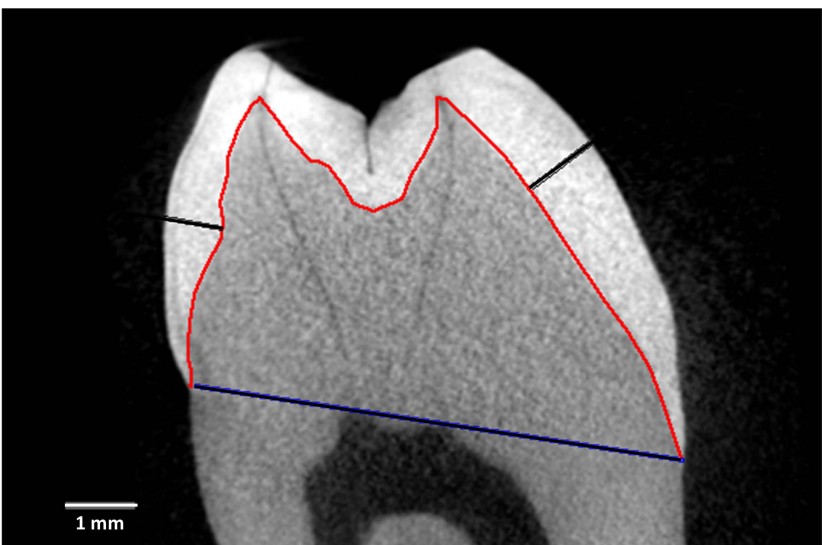

**Figure 1** **Measurement and reference lines in a *Cercocebus atys* lower right third molar.** Specimen is number 94-9b. Image is a virtual slice through the mesial cusps. The red line is the enamel-dentine junction (EDJ) that divides the enamel cap from the dentine. The blue line is the bicervical diameter (BCD). The black lines show maximum lateral enamel thickness in the functional cusp (protoconid, right) and the non-functional cusp (metaconid, left).

measurements were possible for crowns on which dentine horns had been breached. Maximum lateral wall linear measurements were possible with slight wear on cusps.

### P_MF calculation

To calculate estimates of the critical load necessary for a crack to propagate to crown failure, we used the following formula (*Schwartz, McGrosky & Strait, 2020*):

$$P_{MF} = C_F T_e r_e d^{1/2}. \tag{1}$$

Here, $P_{MF}$ is the load at which a margin fracture is estimated to lead to crown failure, $C$ is a constant determined by the elastic moduli of enamel and dentine, $T$ is a constant that is an estimate of the toughness of enamel, $r_e$ is the crown radius, and $d$ is the maximum enamel thickness of the lateral wall (see section 1.2 above). Professor Gary Schwartz (Arizona State University) shared his formula (used in *Schwartz, McGrosky & Strait, 2020*) with us. We used his value of 6 for C and 0.7 for T. As in *Schwartz, McGrosky & Strait (2020)*, maximum lateral wall enamel thickness was assumed to be the ultimate barrier to a crack propagating through the full thickness of the enamel. Also following *Schwartz, McGrosky & Strait (2020)*, $r_e$ was calculated as half of the BCD plus maximum lateral wall thickness on each side. As noted in *Schwartz, McGrosky & Strait (2020)*, this formula provides an approximation of crown strength, which has been validated by experiment, based on modeling the crown as a dome that is loaded at the cusp tip. The data set is included in DataSet S1.
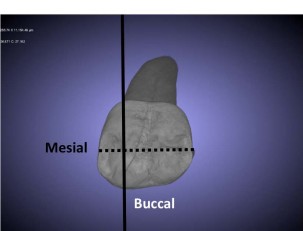
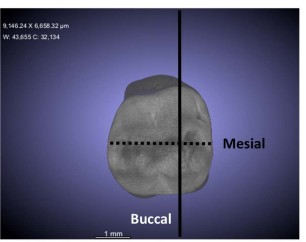
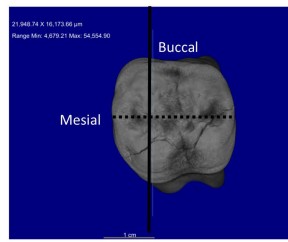

*Cercopithecus cephus* 85-4 ULM2     *Lophocebus albigena* 85-7 ULM2     *Cercocebus atys* 94-7 URM2

**Figure 2** **μCT reconstructions of three molars used in this study.** The dark line shows how molars were virtually sectioned for the analysis of $P_{MF}$. The dotted line shows how these molars were physically sectioned for investigating decussation and nanomechanical properties in trigon basin enamel.

## Statistical analysis

Statistical analyses of $P_{MF}$ were carried out in SAS *v.* 9.4 (SAS Institute, 2015) using Proc Mixed (the Mixed Procedure). This procedure fits mixed linear models to data, allowing a repeated measures analysis of molars belonging to the same individuals and accommodating missing data (*i.e.,* all molar types for upper and lower dentitions were not always available for every individual). Upper molars and lower molars were analyzed separately, and the fixed effects for genus, tooth type and their interaction obtained. Different variance–covariance structures were applied and the model with lowest AIC (Akaike information criterion) was chosen. *Post-hoc* comparison of least-squares means from the regression allowed identification of statistically significant differences between each pair of genera.

## Nanomechanical properties

For analysis of nanomechanical properties, data were collected from three upper second molars–one each–of *Cercocebus atys*, *Lophocebus albigena*, and *Cercopithecus cephus*. Three dimensional μCT reconstructions of these unworn or minimally worn molars are shown in Fig. 2. These molars were physically sectioned along a mesiodistal plane through the length of their trigon basins.

### Nanoindentation procedures

Sections of the three molars were made using a diamond-coated wire saw (STX-202A; MTI Corporation, Richmond, CA, USA). Sections were then mounted in a cold-curing epoxy (Epofix; Streuers, Ballerup, Hovedstaden, Denmark) and polished with a series of SiC papers with progressively finer grits (#800–#4000), finishing with the OP-S polishing suspension on an MD-Dac polishing cloth (Struers, Ballerup, Hovedstaden, Denmark). The mechanical properties of biological tissues are highly dependent on their moisture content, so polished sections were stored for a minimum of 48 h in Hank's balanced salt solution at 4 °C to allow the enamel to rehydrate. Nanoindentation experiments were performed on a Triboindenter nanoindentation system equipped with a diamond Berkovich tip (TI-89;

Bruker, Billerica, MA, USA). Each indent followed a 1.67 mN/s load and unload rate to a maximum load of 5 mN with a hold time of 3 s. A fused quartz reference sample was used to establish the tip area calibration. Indents were performed through the enamel thickness of the trigon basin at six evenly distributed locations, referred to as normalized distance from the EDJ (0 = ~10 µm from the EDJ, 1 = ~10 µm from the outer enamel surface). Hydration was maintained during the tests by smearing a droplet of ethylene glycol on the surface just before beginning indentation. Reduced modulus and hardness values were calculated from the unloading portion of the indentation load–displacement (*Oliver & Pharr, 1992*).

At each location, an array of 25 indents with a spacing of 10 µm between indents was performed (6 locations × 25 indents = 150 indents per tooth). Inner, middle, and outer layers were defined as equal thirds of the enamel thickness. As noted earlier, the co-author who conducted this analysis was blind to the species identity of the molars. The data set is included in DataSet S2.

### Nanoindentation statistical analysis

Hardness, elastic modulus, and elasticity index values were grouped into inner, middle, and outer enamel layers according to their distance from the EDJ. One-way ANOVA testing followed by Tukey *post-hoc* analysis (using python) were used to estimate statistical significance of comparisons among layers and species.

## Enamel decussation

Analysis of enamel decussation was carried out using scanning electron microscopy (SEM) imaging on the three molar sections that were used for nanoindentation, with additional, polishing, etching, and coating steps (see below). Decussation complexity was first assessed through qualitative visual comparison and then quantified in a follow-up analysis.

### Method of quantification

To quantify the complexity of the enamel, we imaged sections using an SEM and measured in-plane and out-of-plane angles of enamel prisms exposed by sectioning. As shown in Fig. 3, if each prism is approximated as a cylinder, it is simple to calculate the in-plane and out-of-plane angles from the elliptical projection of the prism on the imaging plane (*i.e.*, SEM micrograph). Taking the minor radius of the ellipse as equal to the diameter of the prism, (Eq. 2) can be used to relate the minor, $b$, and major, $a$, diameters of the projected ellipse to the angle of intersection with the viewing plane. If a prism is perfectly normal to the viewing plane(out-of-plane angle = 90°), it would present as a circular cross section ($a = b$), while a perfectly in-plane prism (out-of-plane angle = 0°) would have an infinite major diameter ($a = \infty$).

$$\text{out-of-plane angle} = \sin^{-1}\left(\frac{b}{a}\right). \tag{2}$$

### Decussation measurements

Measurements were made manually from SEM micrographs of the central portion of trigon basin (Figs. S1–S3). To improve the contrast in SEM between enamel prisms, a

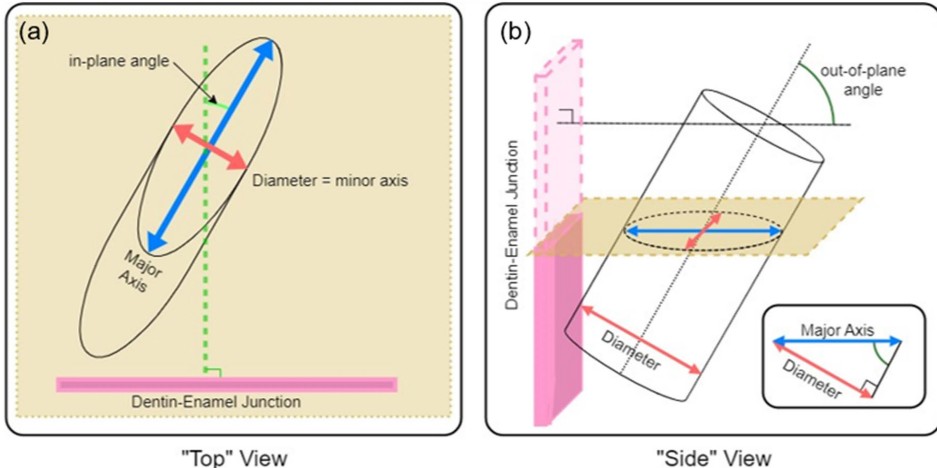

**Figure 3** Schematic description of (A) in-plane and (B) out-of-plane angles derived from the elliptical projection of a cylinder onto a sectioning plane.

two-step etching protocol was developed and applied. First, a 10% HCl etch was applied for 5 s to remove the smear layer from polishing, washed off with copious water, and dried using a lab wipe. This was followed by a 12 s etch with Ultra-Etch 35% phosphoric acid solution (Ultradent), then again rinsed and dried. The enamel was then stored in a vacuum desiccator for >48 h to release any unbound water, followed by 15 nm Pt sputter coating (Leica ACE600) applied to mitigate sample charging. SEM imaging was done on an XL30 system (FEI) with a 5–10 kV accelerating voltage and spot size set to 1–2. Conductive carbon tape was used to fix the sample to the stage and provide a conductive path to ground.

The freehand tool of ImageJ (*Schneider, Rasband & Eliceiri, 2012*) was used to trace the perimeter of a prism and then the major and minor diameters of a fit ellipse that included the entirety of the drawn perimeter was recorded. The in-plane and out-of-plane angle of each prism was then calculated using its major and minor diameters. This process was repeated for at least 50 enamel prisms in each species. The precise prisms measured are documented in Figs. S1–S3.

### Statistical analysis

In-plane and out-of-plane angle distributions for the enamel of each of the three taxa were plotted. For each pair of taxa, distributions were compared using two-sample Kolmogorov–Smirnov (KS) tests (non-parametric), which are sensitive to location, dispersion, and shape.

## RESULTS

*Cercocebus atys* and *Lophocebus albigena* differed in $P_{MF}$ more greatly in functional than non-functional cusps, as hypothesized. Table 1 summarizes the statistical analysis of these differences. While both mangabeys are statistically significantly different from *Cercopithecus*

**Table 1   Repeated measures linear regression results comparing PMF for functional *vs.* non-functional cusps.**

| Upper Molars Functional (Lingual) Cusp | | | | Upper Molars Non-Functional (Buccal) Cusp | | | |
|---|---|---|---|---|---|---|---|
| Effect | df | F value | *p* value | Effect | df | F value | *p* value |
| Genus | 2 | 76.28 | **0.0001** | Genus | 2 | 92.56 | **0.0001** |
| Molar Type | 2 | 8.39 | **0.0027** | Molar Type | 2 | 26.45 | **0.0001** |
| Genus and Molar Type Interaction | 4 | 4.61 | **0.0097** | Genus and Molar Type Interaction | 4 | 9.40 | **0.0003** |

| Upper Molars Least Squares Means Functional (Lingual) Cusp | | | | Upper Molars Least squares Means Non-Functional (Buccal) Cusp | | | |
|---|---|---|---|---|---|---|---|
| Contrast (N) | df | T-value | *p* value | Contrast (N) | df | T-value | *p* value |
| *Cercocebus* ($N = 36$) *vs.* *Cercopithecus* ($N = 18$) | 36 | 12.35 | **0.0001** | *Cercocebus* ($N = 36$) *vs.* *Cercopithecus* ($N = 18$) | 36 | 13.55 | **0.0001** |
| *Cercocebus* ($N = 36$) *vs.* *Lophocebus* ($N = 9$) | 36 | 2.86 | **0.0071** | *Cercocebus* ($N = 36$) *vs.* *Lophocebus* ($N = 9$) | 36 | 1.84 | 0.0735 |
| *Lophocebus* ($N = 9$) *vs.* *Cercopithecus* ($N = 18$) | 36 | 5.26 | **0.0001** | *Lophocebus* ($N = 9$) *vs.* *Cercopithecus* ($N = 18$) | 36 | 6.71 | **0.0001** |

| Lower Molars Functional (Buccal) Cusp | | | | Lower Molars Non-Functional (Lingual) Cusp | | | |
|---|---|---|---|---|---|---|---|
| Effect | df | F value | *p* value | Effect | df | F value | *p* value |
| Genus | 2 | 81.58 | **0.0001** | Genus | 2 | 134.04 | **0.0001** |
| Molar Type | 2 | 17.86 | **0.0216** | Molar Type | 2 | 34.35 | **0.0086** |
| Genus and Molar Type Interaction | 4 | 3.50 | 0.1658 | Genus and Molar Type Interaction | 4 | 4.59 | 0.1205 |

| Lower Molars Least Squares Means Functional (Buccal) Cusp | | | | Lower Molars Least Squares Means Non-Functional (Lingual) Cusp | | | |
|---|---|---|---|---|---|---|---|
| Contrast (N) | df | T-value | *p* value | Contrast (N) | df | T-value | *p* value |
| *Cercocebus* ($N = 11$) *vs.* *Cercopithecus* ($N = 11$) | 19 | 12.23 | **0.0001** | *Cercocebus* ($N = 11$) *vs.* *Cercopithecus* ($N = 11$) | 19 | 15.34 | **0.0001** |
| *Cercocebus* ($N = 11$) *vs.* *Lophocebus* ($N = 9$) | 19 | 2.28 | **0.0341** | *Cercocebus* ($N = 11$) *vs.* *Lophocebus* ($N = 9$) | 19 | 1.65 | 0.1144 |
| *Lophocebus* ($N = 9$) *vs.* *Cercopithecus* ($N = 11$) | 19 | 8.67 | **0.0001** | *Lophocebus* ($N = 9$) *vs.* *Cercopithecus* ($N = 11$) | 19 | 11.73 | **0.0001** |

**Notes.**
Bold indicates statistically significant values at less than $p = 0.05$.

in $P_{MF}$ for functional and non-functional cusps, differences between *Cercocebus atys* and *Lophocebus albigena* are only statistically significant for functional cusps—the protocone of upper molars and protoconid of lower molars. Differences between the two mangabey species are not statistically significant for non-functional cusps in either lower or upper molars. A graphic representation of these differences is given in Fig. 4. Data used in this analysis are available in DataSet S1.

Figure 5 plots data for E (elastic modulus), H (hardness), and the H/E ratio in inner (near the EDJ), middle, and outer (near the tooth surface) thirds of *Cercocebus atys*, *Lophocebus albigena*, and *Cercopithecus cephus* molars. Statistical analysis of nanoindentation data provides strong evidence that *Cercocebus atys* enamel is harder and more wear resistant than that of *Lophocebus albigena* and *Cercopithecus cephus* in most crown regions. This trend is apparent throughout the thickness of the enamel but is most pronounced in the

## Upper Molars

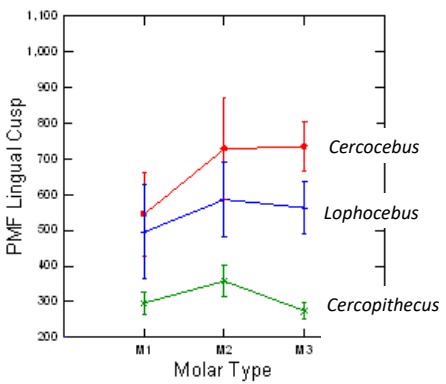 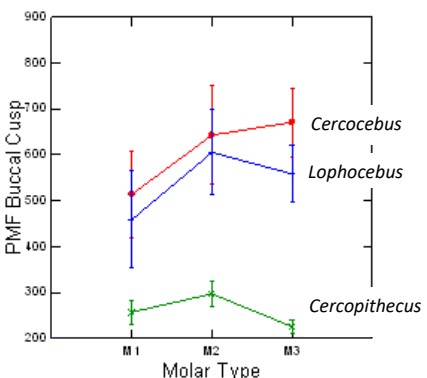

## Lower molars

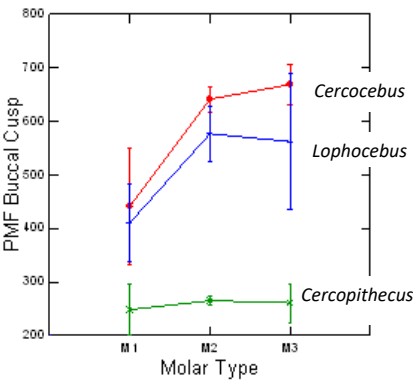 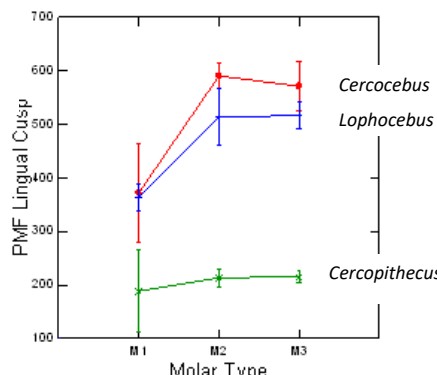

**Figure 4** **Means and 1 SD shown for $P_{MF}$ values in upper and lower molars of *Cercocebus, Cercopithecus*, and *Lophocebus*.** In the upper molars, the lines connecting means for the molars show more separation between *Cercocebus* and *Lophocebus* for the lingual (functional cusp, top left) than they do for the buccal (nonfunctional cusp, top right). In the lower molars, the separation between the *Cercocebus* and *Lophocebus* lines also appears greater for the functional (buccal, bottom left) *vs.* non-functional cusp (lingual, bottom right), but the difference is not as pronounced as it is for the upper molars. Table 1 of the main text contains the results of statistical $P_{MF}$ comparisons.

outermost layer (Fig. 5). Nanoindentation data are available in DataSet S2. *Constantino et al. (2012)* reported primate nanoindentation values that are somewhat greater than ours (by 20–40%), possibly because our study was conducted on hydrated teeth while that of Constantino and colleagues was performed on desiccated samples. Dehydration of enamel leads to increases in both E and H (of approximately 33% on average) (*Huang et al., 2019*), which could explain differences between our two studies.

SEM montages from trigon basins of *Cercocebus atys, Lophocebus albigena,* and *Cercopithecus cephus* are shown in Fig. 6. As expected, molars of all three species had a similar

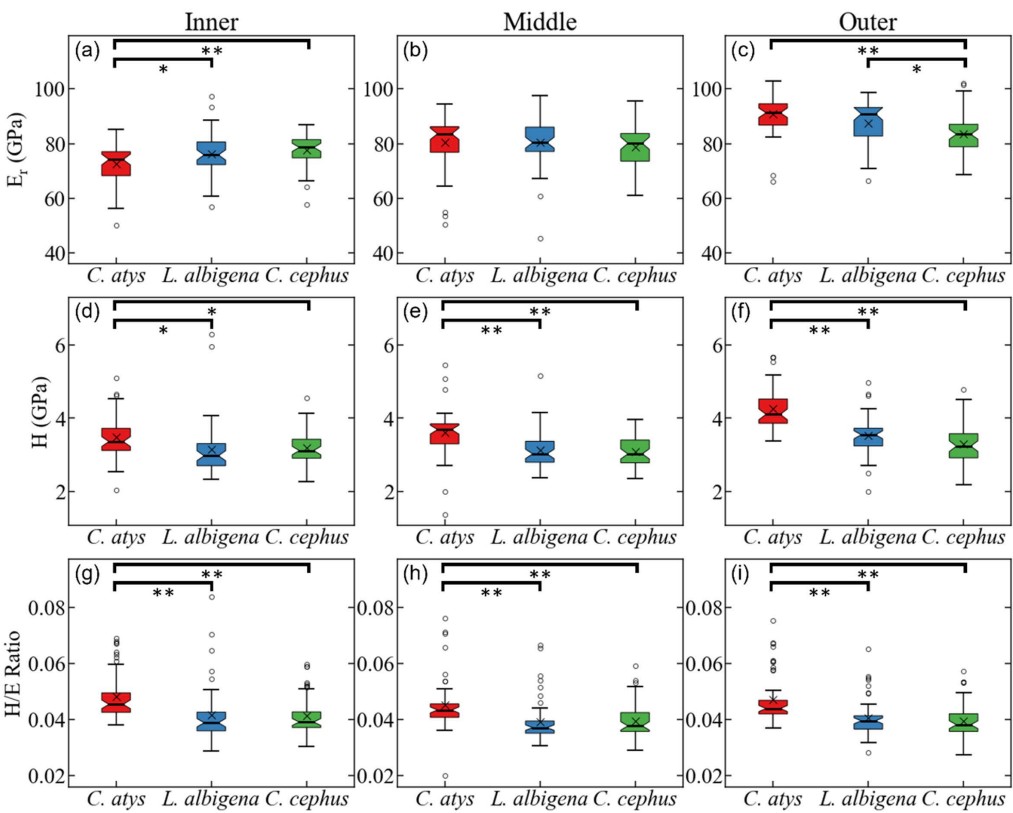

**Figure 5** **Statistical comparison of reduced modulus, hardness, and elasticity index (H/E ratio) from nanoindentation.** Statistical comparison of (A–C) reduced modulus, (D–F) hardness, (G–I) elasticity index (H/E ratio) from nanoindentation of trigon basin enamel separated by layer (A, D, G) inner, (B, E, H) middle, and (C, F, I) outer enamel. Brackets indicate $* p \leq 0.05$ and $** p \leq 0.01$ as quantified by one-way ANOVA and Tukey *post-hoc* analysis.

pattern of parallel prisms near the tooth surface and decussated enamel beneath. Prisms are primarily in-plane ("parazone") in all three molars. However, in the *Cercocebus atys* molar there is a large and isolated bundle of diazone prisms, *i.e.,* a group of transversely-oriented prisms (Fig. 6), that is not present in the other two species' molars. A crack initiating at the outer enamel surface and propagating toward the enamel-dentine junction would be blocked as it intersects this diazone bundle. One such crack appears to have been arrested by this feature in the *Cercocebus atys* molar shown in Fig. 6, though it is not known if this crack occurred pre- or post-mortem.

Quantification of decussation complexity supports this visual assessment. Data used in quantitative comparisons of prism angles are given in DataSet S3. As shown in Fig. 7, enamel in the trigon basin of the *Cercocebus atys* molar has a greater range of prism angles (*i.e.,* exhibits greater complexity) than that of either *Lophocebus albigena* or *Cercopithecus cephus.*

*Cercocebus atys* and *Cercopithecus cephus* both have out-of-plane angles spanning a range of 60° (5–65° and 0–60°, respectively), while *Lophocebus albigena* covers a considerably

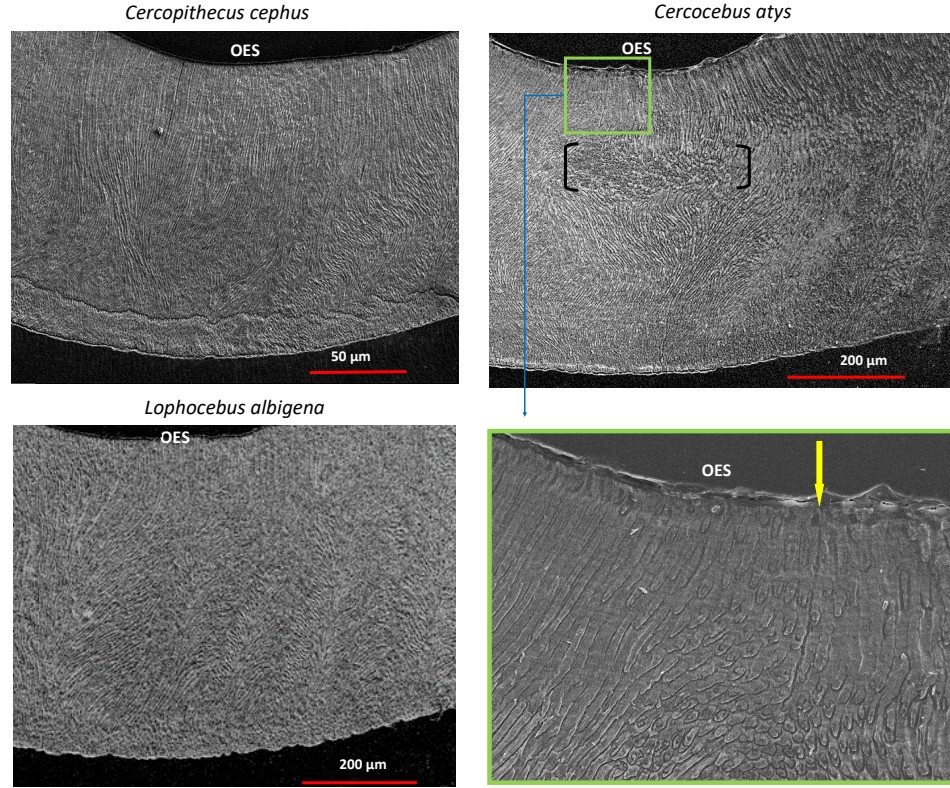

**Figure 6** SEM images of trigon basins of *Cercopithecus cephus*, *Lophocebus albigena*, and *Cercocebus atys*. For *Cercocebus atys*, the black brackets delimit the diazone bundle and the yellow arrow in the enlarged area (green box) points to an enamel crack extending from the OES to the top of the diazone bundle.

smaller span of only 35°. However, a further distinction can be made between *Cercocebus atys* and *Cercopithecus cephus*, as ~18% (9/51) of prisms measured in the former have angles above 55° corresponding to the "bundle" of diazone prisms noted in Fig. 6, while only ~1.6% (2/120) of prisms in the latter fall in the same range. Excluding these two outlying prisms, the distribution of out-of-plane angles in *Cercopithecus cephus* more closely resembles that of *Lophocebus albigena*.

A similar trend is observed in the in-plane measurements, though it is somewhat less pronounced. *Cercocebus atys* enamel shows angles ranging from −60–80° (140° spread) while both *Lophocebus albigena* and *Cercopithecus cephus* range from −40–70° (110° spread). The distributions of angles for both *Lophocebus albigena* and *Cercopithecus cephus* are bimodal, corresponding to the prevailing directions of "bands" visible in Figs. S1–S3.

Kolmogorov–Smirnov tests are summarized in Table 2. All pairwise taxonomic comparisons of in-plane and out-of-plane angle distributions show statistically significant differences from one another.

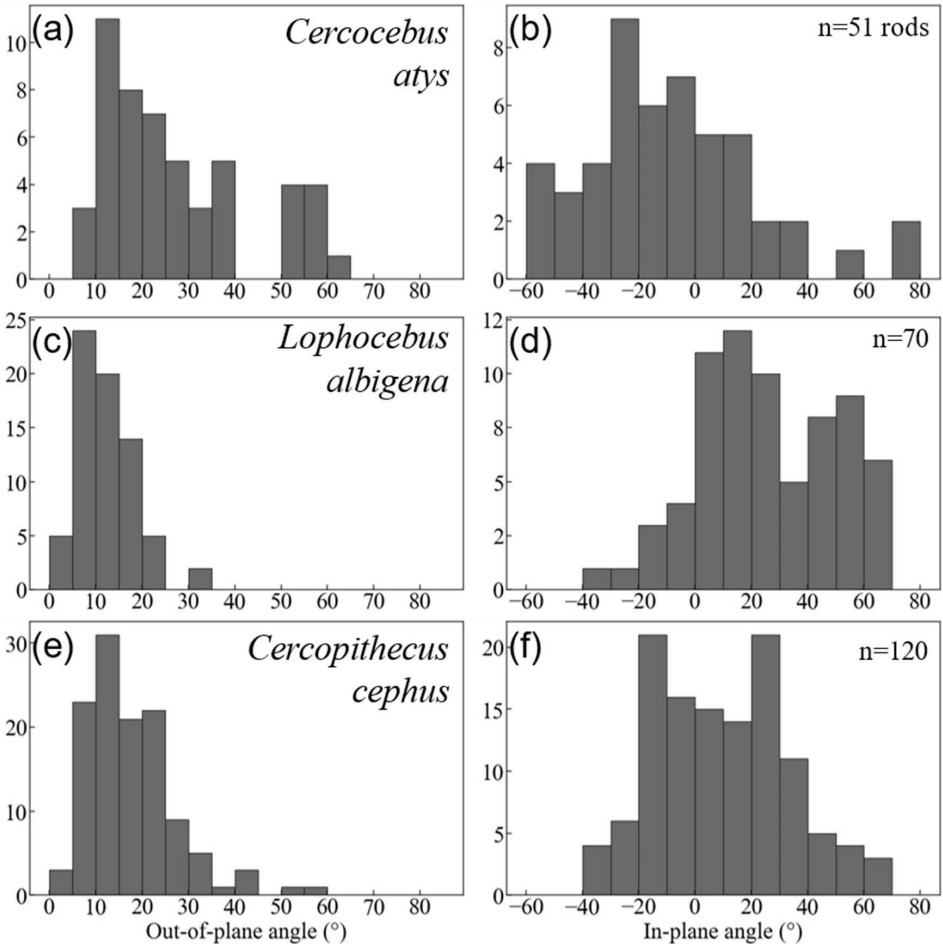

**Figure 7** Histogram summaries of out-of-plane and in-plane angles of prisms in trigon basins of *Cercocebus atys* (A–B), *Lophocebus albigena* (C–D), and *Cercopithecus cephus* (E–F). The total number of prisms measured for each species is reported as n.

**Table 2  Kolmogorov-Smirnov tests for differences in prism angle distributions.**

| Comparison | In-Plane-Angles | | | Out of Plane angles | | |
|---|---|---|---|---|---|---|
| | N | D statistic | *p*-value | N | D statistic | *p*-value |
| *Cercocebus vs.* | 51 | 0.557703 | **0.0001** | 51 | 0.514846 | **0.0001** |
| *Lophocebus* | 70 | | | 70 | | |
| *Cercocebus vs.* | 51 | 0.368137 | **0.0001** | 51 | 0.289706 | **0.0049** |
| *Cercopithecus* | 120 | | | 120 | | |
| *Lophocebus vs.* | 70 | 0.286905 | **0.0014** | 70 | 0.28333 | **0.0017** |
| *Cercopithecus* | 120 | | | 120 | | |

**Notes.**
Bold indicates statistically significant values at less than $p = 0.05$.

## DISCUSSION AND CONCLUSION

The present study addressed the overarching question of whether differences in molar form between *Cercocebus atys* and *Lophocebus albigena* are consistent with known differences between these two species in the frequency with which they eat hard foods. We first hypothesized that differences in the ability of *Cercocebus atys* and *Lophocebus albigena* molar cusps to resist fracture would be most pronounced in their functional cusps. Our estimates of critical loads for margin fractures to propagate to crown failure in *Cercocebus atys vs. Lophocebus albigena* supported this hypothesis. These critical loads were lowest in our comparative sample of *Cercopithecus* species, consistent with their non-durophagous diets. We next hypothesized that in the region of the trigon basin, *Cercocebus atys* would show evidence of greater wear resistance in the form of greater hardness and a greater H/E index than would either *Lophocebus albigena* or *Cercopithecus cephus*. This hypothesis was also supported by our data. Finally, we hypothesized that trigon basin enamel decussation complexity would be greater in the middle and inner enamel of *Cercocebus atys* than it would in these same regions of *Lophocebus albigena* and *Cercopithecus cephus* molars. This hypothesis, too, was supported by our analysis.

Table 3 summarizes aspects of dental form potentially related to dietary differences between *Cercocebus atys* and *Lophocebus albigena* found in this and previous studies. It has long been known that the upper fourth premolars of *Cercocebus* are larger relative to their first molars than those of *Lophocebus* (*Fleagle & McGraw, 1999*). Recent μCT comparison reveals that there are differences between *Cercocebus atys* and *Lophocebus albigena* in the flare of their functional molar cusps (lower molars), absolute crown strength (lower molars), proportional occlusal basin enamel thickness (both upper and lower molars), and relative enamel thickness (lower molars) (*Guatelli-Steinberg et al., 2022*). Some of these features can be seen in Fig. 8. In all cases other than relative enamel thickness, the molars of *Cercocebus atys* are better endowed with features that would resist fracture. We note that absolute crown strength has been argued to provide a more reliable indicator of fracture resistance than relative enamel thickness (*Schwartz, McGrosky & Strait, 2020*). The lower relative enamel thickness of *Cercocebus atys* mandibular molars has been suggested to relate to their greater flare, which increases the dentine core area of these teeth, increasing their size and thus absolute crown strength while driving relative enamel thickness down (*Guatelli-Steinberg et al., 2022*).

To these previous findings, the present study adds that estimated critical loads for margin fractures to propagate to crown failure in *Cercocebus atys'* functional cusps exceed those of *Lophocebus albigena*. It is possible that the greater difference between *Cercocebus atys* and *Lophocebus albigena* in $P_{MF}$ of functional *vs.* nonfunctional cusps is a consequence of how these features scale with molar size, independently of diet. Using previously published data (*Schwartz, McGrosky & Strait, 2020*) and those from the present study, we explored this possibility by graphing $P_{MF}$ of the protoconid and metaconid *vs.* molar size (BCD) both across and within species (Fig. S4). $P_{MF}$ of protoconid and metaconid appear to diverge across species at larger tooth sizes (with BCDs greater than approximately 8.2 mm). It may be that there is a functional explanation for this divergence at larger tooth sizes, if species

**Table 3  Comparison of dental form in *Cercocebus atys* and *Lophocebus albigena* from this and previous studies.**

| Feature | Upper Dentition | Lower Dentition |
|---|---|---|
| Molarization of P4s[a] (size of P4 relative to size of M1) | *C. atys >L. albigena* | *C. atys >L. albigena* |
| Flare of functional molar cusp[b] (lateral wall angle from cemento-enamel junction to cusp tip)[b] | *C. atys ≈ L. albigena* | *C. atys >L. albigena* |
| Relative Enamel Thickness[b] (absolute enamel thickness divided by the square root of the dentine core area) | *C. atys ≈ L. albigena* | *C. atys <L. albigena* |
| Proportional Occlusal Basin Enamel Thickness[b] (occlusal basin enamel thickness relative to average enamel thickness) | *C. atys >L. albigena* | *C. atys >L. albigena* |
| Absolute Crown Strength[b] (A function of crown size and absolute enamel thickness) | *C. atys ≈ L. albigena* | *C. atys >L. albigena* |
| $P_{MF}$ of functional cusp[c] | *C. atys >L. albigena* | *C. atys >L. albigena* |
| $P_{MF}$ of nonfunctional cusp[c] | *C. atys ≈ L. albigena* [c] | *C. atys ≈ L. albigena* |
| Decussation of trigon basin enamel[c] | *C. atys* exhibits more complex enamel | – |
| Hardness, Elastic Modulus, and Elasticity Index of trigon basin enamel[c] | *C. atys >L. albigena* , especially in outer enamel | – |

**Notes.**
[a] *Fleagle & McGraw (1999)*.
[b] *Guatelli-Steinberg et al. (2022)*.
[c] Present study.

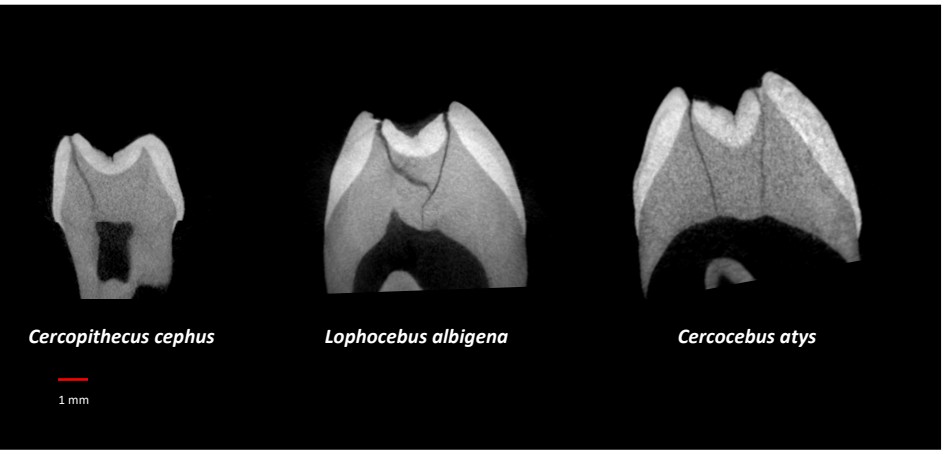

**Figure 8**  μCT virtual slice comparisons among *Cercopithecus cephus* (85-4), *Lophocebus albigena* (85-7), and *Cercocebus atys* (2108). These are upper third molars, oriented with their lingual sides on the right and their buccal sides on the left. Note the thickened occlusal basin enamel in *Cercocebus atys* and its greater lingual cusp flare compared to the upper third molars of the other two species.

with larger teeth (and likely also larger bodies) have more mechanically challenging diets. $P_{MF}$ of the protoconid and metaconid, however, appear to scale with molar size at a similar rate within species—there is at least no marked divergence as there is the cross-species graph. Based on these graphs, there is no clear molar size effect on $P_{MF}$ scaling that is independent of diet, as $P_{MF}$ of the mandibular functional and non-functional cusps appear to scale with molar size at a similar rate within species.

With respect to the nanomechanical properties of trigon basin enamel, the present study revealed that in most enamel regions, *Cercocebus atys* exhibited greater hardness, elastic modulus, and elasticity index values than both *Lophocebus albigena* and *Cercopithecus cephus*, with the most pronounced differences in the outer third of the enamel. *Lophocebus albigena* enamel was significantly different from *Cercopithecus cephus* in elasticity in the outer third of the enamel, but differences between these two species' molars were not statistically significant for hardness and the elasticity index in any enamel region.

In our study, the inner and middle enamel of *Cercocebus atys*' molar trigon basin exhibited greater complexity (*i.e.,* greater heterogeneity in prism orientation) than did these regions of the other two taxa, suggesting greater resistance to crack propagation in *Cercocebus atys*. One of the primary mechanisms of crack growth resistance in enamel is deflection of incident cracks from the plane of maximum opening mode stress as they are guided along the interfaces of adjacent prisms. By forcing the crack along a path of lower opening-mode stress this process mitigates some of the driving force and dissipates energy through the creation of new surfaces (*Yahyazadehfar, Bajaj & Arola, 2013*; *Yang, Bharatiya & Grine, 2022*). Thus, enamel that contains prisms with more tortuous paths would resist cracks more effectively. Comparing trigon basin enamel of *Cercocebus atys* to that of *Lophocebus albigena* and *Cercopithecus cephus*, the wider spread in both in-plane and out-of-plane angles in *Cercocebus atys* would more effectively thwart crack propagation. As noted earlier, there is a trade-off between wear and fracture resistance in the arrangement of enamel prisms, with more decussated enamel being more fracture resistant but less wear resistant than enamel in which prisms are arranged in parallel with the direction of abrasion. Our data suggest that enamel in the trigon basin of *Cercocebus atys* molars gains fracture resistance from its enamel decussation complexity and wear resistance from its nanomechanical properties.

It is important to note that if the enamel were sectioned orthogonally to the presented views, the presentation of the in-plane and-out-plane angles would be reversed. The bands and corresponding double-peak features in the in-plane angles of *Lophocebus albigena* and *Cercopithecus cephus* would likely appear as parazones and diazones similar to those seen in *Cercocebus atys*. However, the differences in in-plane angles between the peaks in *Lophocebus albigena* ($50-20 = 30°$) and *Cercopithecus cephus* ($20-(-20) = 40°$) are somewhat less than the difference of the out-of-plane angles of *Cercocebus atys* ($55-10 = 45°$). Thus, the conclusion that *Cercocebus atys* has the greatest range in decussation angles (both in-plane and out-of-plane) is still supported. *Yang, Bharatiya & Grine (2022)* have recently shown that the frequency of Hunter-Schreger Bands—alternating diazone and parazone bands in enamel—is higher in the functional *vs.* non-functional cusps of human molar enamel (*Daegling, 1992*) . This finding leads us to wonder if the distribution of in-plane and out-of-plane angles vary spatially throughout the enamel in these primate species. Data collection and analysis are currently underway to study this question for a future work.

Taken together, then, the results from the present study and those of previous studies suggest that *Cercocebus atys* molars are both more fracture-resistant and wear-resistant than those of *Lophocebus albigena*. Thus, the two mangabeys, one a routine consumer of

hard foods, the other a fallback hard-object feeder, show differences in molar form in the direction predicted by presumed differences in the frequency with which their molars are subject to the risk of fracture, fatigue stress from cumulative loading, and abrasion.

A further test of the overarching hypothesis of this study—that the frequency of hard-object feeding is associated with differences in molar form—could be achieved by molar form comparisons among other extant species that differ in the frequency with which they consume hard-object foods. Such comparisons are likely to be complicated by several variables, including those associated with oral processing behavior. For example, relative to gracile capuchins (*Cebus* spp.), robust capuchins (*Sapujus* spp.) appear to have craniodental adaptations for hard-object feeding (*Wright, 2005*; *Daegling, 1992*) associated with their greater reliance on hard foods (*Terborgh, 1983*). However, *Sapujus* spp. predominantly use their incisors in ingestive biting, only infrequently biting hard shells with their molars (*Thiery & Sha, 2020*). It is likely for this reason that molar enamel decussation does not differ between *Sapujus* and *Cebus* (*Hogg & Elokda, 2021*), but canine enamel decussation does, with that of *Sapujus* exhibiting greater complexity (*Hogg & Elokda, 2021*). Thus, while it may be possible to identify other comparisons among primates in terms of the frequency with which they consume hard objects and to examine to what extent they are characterized by the suite of features listed in Table 3, such comparisons require knowledge of oral processing behavior and would ideally include comparisons of the hardness of these species' foods.

Many species likely differ in the percentage of hard foods they consume. However, the consumption of hard foods during so-called "fallback" periods is argued to be particularly significant in shaping primate dental and digestive anatomy (*Marshall & Wrangham, 2007*; *Vogel et al., 2008*; *Constantino & Wright, 2009*; *Constantino et al., 2009*; *Marshall et al., 2009*; *Porter, Garber & Nacimento, 2009*; *Yamagiwa & Basabose, 2009*; *Sauther & Cuozzo, 2009*; *Rosenberger, 2013*; *Lambert & Rothman, 2015*). Here we suggest that hard objects consumed both as fallbacks and as dietary staples can shape primate anatomy, and that these two conditions do not give rise to identical molar form. It is of course possible that features associated with greater fracture resistance in *Cercocebus atys* teeth might be exaptations—*i.e.*, they might make possible the regular consumption of hard foods but not have been specifically selected in an evolutionary response to hard food consumption. In either case, given that aspects of molar form can differentiate the two mangabey species with different dietary regimes, it might ultimately be possible to distinguish anatomy associated with regular *versus* fallback hard food consumption in the molar crowns of fossil primates, including hominins.

## ACKNOWLEDGEMENTS

Special thanks to Gary Schwartz for sharing his $P_{MF}$ formula with us.

### Funding

This work was supported by NSF grant 1945008. The funders had no role in study design, data collection and analysis, decision to publish, or preparation of the manuscript.

### Grant Disclosures

The following grant information was disclosed by the authors:
NSF: 1945008.

### Competing Interests

The authors declare they have no competing interests.

### Author Contributions

- Debbie Guatelli-Steinberg conceived and designed the experiments, performed the experiments, analyzed the data, prepared figures and/or tables, authored or reviewed drafts of the article, and approved the final draft.
- Cameron Renteria conceived and designed the experiments, performed the experiments, analyzed the data, prepared figures and/or tables, and approved the final draft.
- Jack R. Grimm conceived and designed the experiments, performed the experiments, analyzed the data, prepared figures and/or tables, and approved the final draft.
- Izabela Maeret Carpenter performed the experiments, prepared figures and/or tables, and approved the final draft.
- Dwayne D. Arola conceived and designed the experiments, authored or reviewed drafts of the article, and approved the final draft.
- W. Scott McGraw conceived and designed the experiments, authored or reviewed drafts of the article, contributed specimens, and approved the final draft.

### Field Study Permissions

The following information was supplied relating to field study approvals (*i.e.*, approving body and any reference numbers):

The remains from Tai Forest are from a field site. Permits from Cote D'Ivoire: CIS2016-03-035,

CIS2010-06-015, CIS2004-01-02

### Data Availability

The raw data is available in the Supplemental Files.

### Supplemental Information

Supplemental information for this article can be found online at http://dx.doi.org/10.7717/peerj.16534#supplemental-information.

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
