# Peer review of "How mangabey molar form differs under routine vs. fallback hard-object feeding regimes"

_PeerJ, doi:10.7717/peerj.16534_

## Round 0.1 · original submission · Minor Revisions

Thanks for submitted to PeerJ. All reviewers had favorable views of your manuscript. Reviewers 2 and 3 suggest some areas and topics that may require further clarification. Please respond to their comments, making changes where you think fit and justifying otherwise. I will read over your revision and make a final decision.

·

Basic reporting

This paper presents a relatively straightforward question, but one which has been important to assess: that is, the degree to which frequency of feeding on mechanically challenging foods affects dental microanatomy and biomechanics.

The question and hypotheses are clearly stated, and the rationale well-explained, as are the methods and results. I see no major concerns with basic reporting in this paper.

Experimental design

The design is appropriate to the question, and I am particularly satisfied that the authors used both mechanical testing and histological data. I am not super-qualified to comment on some of the biomechanical methods, but they appear to be in line with best practice.

Validity of the findings

The results appear accurate and well-reported, and the authors’ interpretations are supported by the data.

Additional comments

This is a well-presented study and is worthy of publication.

My (minor) piece of constructive criticism is that it would be good to address in the introduction the fact that that different schmeltzmusters have been interpreted as a tradeoff between fracture resistance and wear, and how that affects your hypotheses and methodology, since you are predicting that Cercocebus will be better at both. Somewhere around line 130-ish would seem to fit.

Reviewer 2 ·

Basic reporting

This manuscript meets all the criteria defined for "Basic Reporting."

Experimental design

Strengths
This is an interesting and original comparison between two Old World primates that ingest mechanically challenging foods. Identifying differences in anatomy between a species that habitually consumes such foods and one that is more of an intermittent consumer of such foods has implications for understanding both modern and fossil species.

The hypothesis in question is well defined. While I don't have the technical knowledge to critique the enamel analysis in detail, it appears to be very thorough and includes pertinent variables pertaining to enamel strength and toughness. The methods could be replicated.

Weaknesses
I think previous work concerning the diet of these species needs to be used to explain the focus on the molar teeth in the analysis. It seems mastication is the focus, but overcoming tissues that are not ultimately masticated may be the key to exploiting such hard foods. Please explain how these species ingest and masticate the hard foods. Do they use their hands? Do they initially bite with anterior dentition and/or premolars? What are toughness and stiffness (elastic modulus) values for bitten and chewed tissues? Being able to answer these questions may better define the adaptations needed for fallback feeding versus habitual hard food feeding. By focusing on the molars it seems only the evolutionary implications of ingested tissues are investigated. I also wonder if food tissue toughness may really be the differentiating property here.

Validity of the findings

In terms of validity of findings, I feel the criteria are met overall. This said, I think it is important to make the point that masticated tissues are the focus, unless ingestive biting occurs on the molars.

Additional comments

NA

·

Basic reporting

no comment

Experimental design

no comment

Validity of the findings

no comment

Additional comments

Thank you for asking me to review the manuscript “How mangabey molar form differs under routine vs. fallback hard-object feeding regimes” by Guatelli-Steinberg et al. Overall I found this paper to be well written, the research interesting, and the results well justified by the data. Thanks to the authors for generating this interesting work.

My primary comments mostly relate to ease of reading and clarity of the paper. Because of the number of analyses and variables analyzed I often found it hard to follow exactly what was being discussed. In addition, there were multiple instances where abbreviations were provided but not defined (e.g., AET, HSB, OES) or where abbreviations were inconsistently used. Some of these can be easily explained in the text, but I think one way to improve clarity of the paper would be to add a table with all of the variables analyzed, their abbreviations, definitions, and what you expect to observe among the species in the sample. This could easily link back to the hypotheses outlined in the introduction and I think would make it a lot easier to see the overall scope of the analyses.

In a similar vein, the discussion as currently organized is a bit hard to follow. It would be nice if, at the very start, the authors returned to and restated their overarching research question and hypotheses before jumping into the results of prior research as they currently do. I also would have liked to see the authors more clearly support or refute their hypotheses as they are laid out in the introduction; as written now the text does return to each of their questions, but never clearly says whether their hypotheses were supported. Perhaps some additional organization/ subheaders would be valuable for the discussion in this regard.

Some additional larger context at the end of the discussion would also be useful. For example, do the authors believe these results will be broadly generalizable to other species? Are there any other pairs of extant taxa where this might be useful to look at? What do other data on enamel form for primates suggest here? The authors end by hinting at relevance for hominins and other fossil primates, but exactly how might this be implemented? Could this be expanded upon a bit? I think adding more to this section would make this paper more broadly applicable.

There were a few other smaller items I noted during my review:
--Line 108- change to “crushing basin of the upper molars”
--Line 111- change to “and the ratio between the two (H/E), known…”
--Line 313- should this be referring to Figure 6 instead of Figure 3?
--Figures 4 and 5- it would be really nice if the color schemes for the species were the same in each of these figures. This seems small but it would add a bit of visual continuity that would make the images easier to interpret more quickly
--Figure 6- can the authors please add a key to the images to show which surfaces are which? I assume the occlusal surface is at the top of the images but it’s a bit hard to tell

---

## Round 0.2 · accepted · Accept

Thank you for thoroughly addressing the reviewers' comments. Most of their comments broadly concerned the organization and flow of the manuscript, including requests to provide some more information on background in the Introduction (Reviewers 1 and 2) and to discuss comparisons with other primate radiations in the Discussion. In all cases, you've done a great job of addressing these points within the limits of existing knowledge, and otherwise remaining appropriately circumspect. Overall, I have assessed the changes you made in response to these and other (more minor) concerns, and consider the manuscript now ready for publication. Once again, thanks for submitting this research to PeerJ.